# Enhancement of properties in Mizar

Artur Korniłowicz

Institute of Computer Science, University of Bialystok, Bialystok, Poland

## ABSTRACT

A "property" in the Mizar proof-assistant is a construction that can be used to register chosen features of predicates (e.g., "reflexivity", "symmetry"), operations (e.g., "involutiveness", "commutativity") and types (e.g., "sethoodness") declared at the definition stage. The current implementation of Mizar allows using properties for notions with a specific number of visible arguments (e.g., reflexivity for a predicate with two visible arguments and involutiveness for an operation with just one visible argument). In this paper we investigate a more general approach to overcome these limitations. We propose an extension of the Mizar language and a corresponding enhancement of the Mizar proof-checker which allow declaring properties of notions of arbitrary arity with respect to explicitly indicated arguments. Moreover, we introduce a new property—the "fixedpoint-free" property of unary operations—meaning that the result of applying the operation to its argument always differs from the argument. Results of tests conducted on the Mizar Mathematical Library are presented.

## INTRODUCTION

Classical mathematical papers consist of sequences of definitions and justified facts classified into several categories like: theorems, lemmas, corollaries, and so on, often interspersed with some examples and descriptions.

Mathematical documents prepared using various proof-assistants (e.g., Isabelle (*Isabelle, 2020*), HOL Light (*HOL Light, 2020*), Coq (*Coq, 2020*), Metamath (*Metamath, 2020*), Lean (*Lean, 2020*), and Mizar (*Mizar, 2020*)) can also contain other constructions that are processable by dedicated software. In the case of the Mizar system (*Bancerek et al., 2015*; *Grabowski, Korniłowicz & Naumowicz, 2010*) such constructions are:

1. *existential*, *conditional* and *functorial registrations* which enhance processing adjectives (*Naumowicz, 2009*),
2. *term reductions* which reduce terms to their proper sub-terms (*Korniłowicz, 2013*),
3. *term identifications* which identify equivalent notions from different theories (*Grabowski, Korniłowicz & Naumowicz, 2010*), and
4. *properties* which can declare chosen properties of predicates, functors and types at the stage of their definitions (*Naumowicz & Byliński, 2004*).

The current implementation of the Mizar proof-assistant allows using properties for notions with a specific number of visible arguments. Visible arguments are those which are explicitly used in the notation of the notion. For example, if $x$ and $y$ are elements of

Corresponding author
Artur Korniłowicz,
arturk@math.uwb.edu.pl

a group $G$, then for the operation $x + y$, where $+$ denotes the addition of elements of the group $G$, $x$ and $y$ are visible arguments, while $G$ is a hidden argument of the operation.

In this paper we propose an extension of both the Mizar language and the Mizar proof-checker which allows declaring properties of notions of arbitrary arity with respect to explicitly indicated arguments. We also introduce a new property—the "fixedpoint-free" property of unary operations. It states that the result of applying the operation to its argument always differs from the argument.

The structure of the paper is the following: in 'Mizar proof-assistant' we present the Mizar proof-assistant with the focus on its features related to the new development proposed in this paper; in 'Properties' we describe how to define and use properties for arbitrary arguments; in 'Fixedpoint-free Property' we present the "fixedpoint-free" property; and finally, in 'Conclusions and Future Work', we describe some conclusions and plans for next enhancements of properties in Mizar. The results of implementing new features in the Mizar Mathematical Library (MML) (*Bancerek et al., 2018*; *Alama et al., 2011*) are shown in both 'Properties' and 'Fixedpoint-free Property'.

## MIZAR PROOF-ASSISTANT

The Mizar project started in 1973 under the leadership of Andrzej Trybulec (*Matuszewski & Rudnicki, 2005*; *Grabowski, Korniłowicz & Naumowicz, 2015*). The main goal of the project is to develop a computer framework that allows writing mathematical papers under the control of computer programs that check syntactical, semantical and logical correctness of texts. The Mizar project consists of three main components:

- a language invented to write mathematical texts to be processed by computers,
- a collection of computer programs designed and implemented for processing texts written in the Mizar language, with its core program, a proof-checker named VERIFIER, suitable for formal verification (*Avigad & Harrison, 2014*; *Trybulec et al., 2013*; *Wiedijk , 2006*), and
- the Mizar Mathematical Library—a library of documents (called articles) written in the Mizar language and verified by the Mizar proof-checker.

### Language
The Mizar language reflects the natural language of mathematics and enables computers to efficiently process documents written in the language. It implements rules for writing: formulae of various kinds, definitions, theorems, local lemmas, reasoning methods, proof steps, and other syntactic constructions instructing the proof-checker to launch dedicated algorithms for processing particular mechanisms (e.g., term identifications (*Grabowski, Korniłowicz & Naumowicz, 2010*), term reductions (*Korniłowicz, 2013*), properties of predicates and functors (*Naumowicz & Byliński, 2004*)), etc.

For the purposes of this paper we recall some basic information about how new mathematical notions can be defined in Mizar articles.

The Mizar language allows users to define predicates, functors (linguistic functions used to define operations), attributes (*Naumowicz, 2009*), types (*Bancerek, 2003*), and

structures. The general form of a definition consists of the definition arguments, permissive assumptions (necessary to prove the correctness of the definition), its notation (prefix, infix or suffix), the result type, the definiens, correctness conditions, and some extra properties. For example, the union of two sets can be defined as follows:

```
definition
  let X,Y be set;
  func X \/ Y -> set means
  :: XBOOLE_0:def 3
  for x being object holds x in it iff x in X or x in Y;
  existence;
  uniqueness;
  commutativity;
  idempotence;
end;
```

where the statement `let X,Y be set;` introduces two arguments, `func` declares that it is a definition of an operation, `X \/ Y` introduces the symbol `\/` for the union and declares it to be used as an infix symbol, $\rightarrow$ `set` defines the type of the result of the operation (the union of two sets is a set), `XBOOLE_0:def 3` is a unique identifier of the definition (it can be used to refer to the definition), `for` statement describes the meaning of the definition (`it` keyword in the definiens represents the notion being defined), `existence` is an automatically generated statement that has to be proved by authors to justify that there exists an object satisfying the definiens, `uniqueness` is another automatically generated statement that has to be proved to justify that there exists only one object satisfying the definiens, `commutativity` and `idempotence` are extra properties that can be declared and proved about the notion at this stage.

One can observe that in the above example definition there are no permissive assumptions, because they are not necessary to justify the existence and uniqueness. But, for example, in the definition of a homeomorphism between two topological structures:

```
definition
  let S,T be TopStruct;
  assume S,T are_homeomorphic;
  mode Homeomorphism of S,T -> Function of S,T means
  :: TOPGRP_1:def 3
  it is being_homeomorphism;
  existence;
end;
```

such an assumption `assume S,T are_homeomorphic;` is necessary to justify the existence, because in general not all topological spaces are homeomorphic.

The Mizar language allows also introducing, so called, *redefinitions*. Redefinitions can be used for the following purposes:

- to change result types of operations for more specific types of arguments; for example, addition of numbers can be defined for complex numbers with the result type

representing complex numbers, but when arguments of the definition are, say, natural numbers, the result type of the addition can be redefined to the type representing natural numbers;

- to reformulate definiens formulae in domain languages adequate to the types of arguments of the notion; for example, the inclusion of arbitrary sets can be defined in terms of elements of the sets, while when arguments of the inclusion are binary relations, the definiens of the inclusion can be formulated in terms of pairs of elements.

## Proof checker

The logical foundation of the Mizar checker is classical first-order logic with equality (in some contexts, however, free second-order variables are permitted enabling the introduction of schemes, e.g., the scheme of mathematical induction). The proof system is based on the Jaśkowski style of natural deduction (*Jaśkowski, 1934*). Structures of proofs are basically related to the structures of the formulae being proved with application of definitional expansions.

From the author's perspective, the correctness of formalized reasoning is controlled by the core utility of the Mizar system, the VERIFIER. Although its proof-checking code is sufficient to guarantee logical correctness, there are successful applications of external software to perform some particular tasks during processing Mizar texts (*Naumowicz, 2015*; *Naumowicz, 2014*; *Naumowicz, 2010*).

VERIFIER is a classical proof checker based on the notion of the inference obviousness (*Davis, 1981*; *Rudnicki, 1987*). The basic modules of VERIFIER are the following:

| | |
|---|---|
| PARSER | which is responsible for controlling the lexical structure of a given text and generating the parse tree of the text. |
| MSM PROCESSOR | which identifies constants, variables and labels. |
| ANALYZER | which identifies objects and operations, performs type checking and resolves possible ambiguities caused by overloading of symbols. Moreover, it verifies if constraints required by particular constructions are fulfilled. |
| REASONER | which controls structures of proofs according to the natural deduction rules. |
| CHECKER | which verifies logical correctness of inferences. As a disprover it tries to refute negations of processed sentences. It performs propositional calculus (PRECHECKER), equational calculus over equalities accessible in inferences (EQUALIZER) and unification (UNIFIER). |

## Mizar Mathematical Library

The Mizar Mathematical Library (MML) (*Bancerek et al., 2018*; *Alama et al., 2011*) was established in 1989 to accumulate mathematical knowledge formalized and verified using the Mizar proof-assistant. It is a collection of papers based on the Tarski-Grothendieck (TG) set theory, which is a variant of the ZFC set theory (*Hayden et al., 1968*), where the axiom of infinity is replaced by Tarski's axiom of existence of arbitrarily large, strongly inaccessible cardinals (*Tarski, 1939*).

The current Version 5.63.1382 of the Mizar Mathematical Library contains 1385 articles (107,756,687 bytes in total) devoted to various branches of mathematics.

Developing the MML includes the following tasks:

- collecting new knowledge realized by: (a) developing background knowledge to prepare a comprehensive database for practicing mathematicians and for educational purposes; (b) formalizing entire mathematical books (*Gierz et al., 1980*; *Bancerek & Rudnicki, 2002*); (c) formalizing well-known theorems (*Abad & Abad, 1999*); and (d) developing new theories (*Grabowski, 2014*; *Grabowski, 2013*; *Grabowski & Jastrzebska, 2010*).
- refactoring the database (*Grabowski & Schwarzweller, 2007*) to keep its integrity (*Rudnicki & Trybulec, 2003*) and to increase readability of the stored proofs (*Pąk, 2014*).

Knowledge stored in the database is used in various branches of science and education, e.g., for representing mathematics on WWW (*Iancu et al., 2013*; *Urban, 2005*), as an input for ATP systems (*Urban, Rudnicki & Sutcliffe, 2013*; *Urban & Sutcliffe, 2010*; *Urban, Hoder & Voronkov, 2010*; *Urban, 2008*), as an input for services classifying mathematics (*Grabowski & Schwarzweller, 2012*), and others.

## Processing Mizar articles

Every Mizar article written as a plain text file with the file extension `.miz` consists of two main parts: its *environment*, which can be seen as the import part from the Mizar Mathematical Library, and *text-proper* part, where new definitions, lemmas, theorems etc. are placed.

In the environment part the following directives are allowed:

- `vocabularies` –imports symbols of notions stored in the MML.
- `notations` –imports notations of notions stored in the MML. The order is important –in the case of overloading the last one counts.
- `constructors` –imports constructors (meanings) of notions.
- `theorems` –imports theorems to which proofs refer to.
- `schemes` –imports schemes to which proofs refer to.
- `definitions` –imports formulae that determine proof skeletons.
- `registrations` –imports registrations, term identifications and term reductions used in proofs.
- `equalities` –imports equalities of operations defined using `equals` clause with their meanings.
- `expansions` –imports expansions of predicates and adjectives.
- `requirements` –imports *switches* to launch build-in procedures by the checker.

The environment is processed by a dedicated program—ACCOMMODATOR. It reads the environment part of the article and prepares global notions ready to be used in the local article. When it is done, VERIFIER processes the text-proper part of the article. Firstly, PARSER scans the article, checks its grammatical correctness and prepares the parse tree of the article. The parse tree is stored in the XML file with the extension `.wsx` (*Naumowicz & Piliszek, 2016*). The next submodule, MSM PROCESSOR, reads the `.wsx` file and identifies

all identifiers of constants, variables and labels that appear in the article. MSM PROCESSOR adds computed information to data written in the `.wsx` file and creates another XML file with the extension `.msx`. Then, ANALYZER reads the `.msx` file and resolves ambiguities and identifies used notions (predicates, adjectives, types, operations and structures). ANALYZER creates another XML file with the extension `.xml`—the structure of this `.xml` file differs from structures of both `.wsx` and `.msx` files. The `.xml` file contains the complete semantic information about all constructions used in the processed article. When all ambiguities are resolved and all notions are identified, the article is ready to be verified against its logical correctness by the Mizar checker. Formulae are negated, transformed to their disjunctive normal forms and all disjuncts, one by one, are then verified by EQUALIZER—a Mizar module dealing with equational calculus (*Rudnicki & Trybulec, 2001*). It collects all terms from the processed disjunct, and computes the congruence closure over equalities available in the inference. The equalities can be provided by various Mizar constructions, like: term expansions (`equals`), properties of operations, term reductions, term identifications, arithmetic, type changing (`reconsider`), and others, e.g., processing structures.

For the sake of this paper let us underline properties of operations. They are described in more detail in 'Properties'.

The last procedure applied to the processed inference is its unification. If EQUALIZER cannot disprove the formula, UNIFIER starts working and tries to refute it. If UNIFIER finds a contradiction, the original disjunct is accepted as true; otherwise, appropriate messages are reported and authors are supposed to complete missing proofs.

When all formulae are accepted, the article can be submitted to the Mizar Mathematical Library and the new knowledge can be used in subsequent works. Two other tools are used to export the new article into the database: EXPORTER—extracts public knowledge from the article, and TRANSFERER—transfers the knowledge into the Mizar Mathematical Library.

## PROPERTIES

Properties in Mizar are constructions which can be used to declare that predicates are reflexive ($\forall_x : xRx$), irreflexive ($\forall_x : \neg xRx$), symmetric ($\forall_{x,y} : xRy \rightarrow yRx$), asymmetric ($\forall_{x,y} : xRy \rightarrow \neg yRx$), and connected ($\forall_{x,y} : xRy \vee yRx$); in the case of operations, they can be declared as involutive ($f(f(x)) = x$), projective ($f(f(x)) = f(x)$), idempotent ($f(x,x) = x$), and commutative ($f(x,y) = f(y,x)$). Such declarations of chosen properties must be placed within *definitional blocks*. When a notion is equipped with some properties, then adequate formulae involving the notion become obvious to the Mizar checker without any explicit reference to the definition and any theorem (they are processed automatically based on internally generated equalities of terms in cases of properties of functors and appropriate formulae in cases of properties of predicates). For example, the declaration of the idempotence of the binary union of sets implies that the equality $A \cup A = A$ becomes obvious for any set $A$.

The current implementation of the Mizar checker is restricted to fixed numbers of visible arguments of considered notions listed in Table 1.

In this work we propose an extension of the Mizar system with the possibility of explicit indication with respect to which visible arguments of mathematical notions given properties

**Table 1   Arities of properties of notions.**

| Predicates | | Functors | |
|---|---|---|---|
| **Property name** | **Arity** | **Property name** | **Arity** |
| Reflexivity | 2 | Projectivity | 1 |
| Irreflexivity | 2 | Involutiveness | 1 |
| Symmetry | 2 | Commutativity | 2 |
| Asymmetry | 2 | Idempotence | 2 |
| Connectedness | 2 | | |

can be declared. To achieve this, one can use the `wrt` clause followed by a comma separated `list_of_loci` of visible arguments of lengths presented in Table 1. The extended syntax of a definition of a functor is the following:

> **definition**
> **let** $x_1$ **be** $\theta_1$, $x_2$ **be** $\theta_2$, …, $x_n$ **be** $\theta_n$;
> **func** $\otimes(x_1, x_2, …, x_n)$ -> $\theta$ **means** :*ident*:
>  $\Phi(x_1, x_2, …, x_n, \textbf{it})$;
> **correctness;**
> **property_name wrt list_of_loci justification;**
> **end;**

and the extended syntax of a definition of a predicate is the following:

> **definition**
> **let** $x_1$ **be** $\theta_1$, $x_2$ **be** $\theta_2$, …, $x_n$ **be** $\theta_n$;
> **pred** $\pi(x_1, x_2, …, x_n)$ **means** :*ident*:
>  $\Phi(x_1, x_2, …, x_n)$;
> **property_name wrt list_of_loci justification;**
> **end;**

For the back compatibility the `wrt` clause is not obligatory, definitions with no `wrt` clause work as in previous releases of the Mizar checker.

## Examples

As an example of using this new feature in the MML we can cite the theorem (*Kusak & Radziszewski, 1991*):

```
theorem Th78:
  sum(a,b,o) = sum(b,a,o);
```

which can be reformulated as commutativity of:

```
definition
  let SAS be Semi_Affine_Space;
  let a,b,o be Element of SAS;
  func sum(a,b,o) -> Element of SAS means :Def5:
  congr o,a,b,it;
  correctness by Th62,Th63;
  commutativity wrt a,b by Th69;
```

```
end;
```

An interesting example of a theorem which at the first glance looks like symmetry of some quaternary predicate with respect to the third and forth argument, but cannot be reformulated as symmetry of the predicate, is *Oryszczyszyn & Prazmowski (1990)*:

```
theorem Th24:
  p,q _|_ p1,q1 implies p,q _|_ q1,p1;
```

To explain this fact one should look at types of variables p, q, p1, and q1 used in the theorem and types of arguments of the definition of the predicate _|_. The type of p, q, p1, and q1 is

```
reserve V for RealLinearSpace;
reserve w,y for VECTOR of V;
reserve p,p1,q,q1 for Element of AMSpace(V,w,y);
```

while the predicate _|_ is defined as:

```
definition
  let POS be OrtStr;
  let a,b,c,d be Element of POS;
  pred a,b _|_ c,d means
  :: ANALMETR:def 5
  [[a,b],[c,d]] in the orthogonality of POS;
end;
```

Now it is clear that types of variables p, q, p1, and q1 are more restricted than original types of arguments a, b, c, and d of the predicate declared in its definition. The statement proved as the mentioned theorem Th24 is true for elements of a particular space AMSpace(V,w,y), but does not hold for elements of an arbitrary space OrtStr.

It is a very typical case, when some notion is defined for general types of arguments, and its particular properties are provable for less general ones.

## Changes in XML files

As it was said in 'Processing Mizar articles', the Mizar verifier, to check the correctness of Mizar articles, generates and processes several intermediate files written in XML formats. To be able to implement the feature considered in this section, we had to slightly change the grammars of these XML files. From the perspective of Mizar users formalizing some knowledge, these changes are not important—the authors are not supposed to look into these files. For researchers who use the Mizar system for other purposes and develop external applications working on the semantic level of the Mizar Mathematical Library (*Urban, 2005*), these changes will induce the need for some adjustments or reimplementations. Therefore, below we explain the changes. Let us take the following definition:

```
definition
  let E be set;
  let A,B be Element of E;
```

```
  func +(A,B,E) -> Element of E equals
  A \/ B;
  coherence;
  commutativity wrt A,B;
end;
```

as a working example.

In the file `.wsx` (*Naumowicz & Piliszek, 2016*) we added a new XML element `PropertyLoci` including a list of loci for which the property holds. For our example it looks like this:

```
<Item kind="Property" property="commutativity" line="15" col="15"
      posline="15" poscol="23">
 <PropertyLoci>
  <Locus idnr="5" spelling="A" line="15" col="21"/>
  <Locus idnr="6" spelling="B" line="15" col="23"/>
 </PropertyLoci>
 <Straightforward-Justification line="15" col="23"/>
</Item>
```

This is propagated to the `.msx` file, which is an extension of the `.wsx` file, and for our example it becomes:

```
<Item kind="Property" property="commutativity" line="15" col="15"
      posline="15" poscol="23">
 <PropertyLoci>
  <Locus idnr="5" spelling="A" line="15" col="21" origin="Constant"
         kind="Constant" serialnr="2" varnr="2"/>
  <Locus idnr="6" spelling="B" line="15" col="23" origin="Constant"
         kind="Constant" serialnr="3" varnr="3"/>
 </PropertyLoci>
 <Straightforward-Justification line="15" col="23"/>
</Item>
```

Next two changes are introduced in `.xml` files: we added internal descriptions of properties in their definitions:

```
<JustifiedProperty>
<Commutativity>
<PropertyLoci>
<Int x="2"/>
<Int x="3"/>
</PropertyLoci>
</Commutativity>
```

and in constructors with which properties are associated:

```
<Constructor kind="K" nr="1" aid="EE" relnr="20">
```

```
<Properties>
<Commutativity propertyarg1="2" propertyarg2="3"/>
</Properties>
```

## FIXEDPOINT-FREE PROPERTY

As another enhancement of properties in Mizar we propose a new unary property of functors—"fixedpoint-free". The `fixedpoint-free` property is meaningful for operations of which the result of application to a given argument is always different from the argument. This is reflected in justification formulae of the properties to be proved at the stage of defining the operation.

We propose the following syntax and formulae to be proved to justify the correctness of the property for given functors. In the case of functors defined using the `means` clause with a simple definiens it is:

**definition**
  **let** $x_1$ **be** $\theta_1$, $x_2$ **be** $\theta_2$, …, $x_n$ **be** $\theta_n$;
  **func** $\otimes(x_n)$ -> $\theta_{n+1}$ **means** *:ident:* $\Phi(x_1, x_2, \ldots, x_n, \textbf{it})$;
  **existence;**
  **uniqueness;**
  **fixedpoint-free**
  **proof**
    **thus for** $r$ **being** $\theta_{n+1}$, $x$ **being** $\theta_n$ **st** $\Phi(x_1, x_2, \ldots, x_{n-1}, x, r)$
      **holds** $r <> x$;
  **end;**
**end;**

using the `means` clause with a complex definiens it is:

**definition**
  **let** $x_1$ **be** $\theta_1$, $x_2$ **be** $\theta_2$, …, $x_n$ **be** $\theta_n$;
  **func** $\otimes(x_n)$ -> $\theta_{n+1}$ **means** *:ident:*
    $\Phi_1(x_1, x_2, \ldots, x_n, \textbf{it})$ **if** $\Gamma_1(x_1, x_2, \ldots, x_n)$,
    $\Phi_2(x_1, x_2, \ldots, x_n, \textbf{it})$ **if** $\Gamma_2(x_1, x_2, \ldots, x_n)$,
    $\Phi_3(x_1, x_2, \ldots, x_n, \textbf{it})$ **if** $\Gamma_3(x_1, x_2, \ldots, x_n)$
    **otherwise** $\Phi_n(x_1, x_2, \ldots, x_n, \textbf{it})$;
  **existence;**
  **uniqueness;**
  **consistency;**
  **fixedpoint-free**
  **proof**
    **thus for** $r$ **being** $\theta_{n+1}$, $x$ **being** $\theta_n$ **st**
      $(\Gamma_1(x_1, x_2, \ldots, x_n)$ **implies** $\Phi_1(x_1, x_2, \ldots, x_{n-1}, x, r))$ **&**
      $(\Gamma_2(x_1, x_2, \ldots, x_n)$ **implies** $\Phi_2(x_1, x_2, \ldots, x_{n-1}, x, r))$ **&**
      $(\Gamma_3(x_1, x_2, \ldots, x_n)$ **implies** $\Phi_3(x_1, x_2, \ldots, x_{n-1}, x, r))$ **&**
      $(\textbf{not } \Gamma_1(x_1, x_2, \ldots, x_n)$ **& not** $\Gamma_2(x_1, x_2, \ldots, x_n)$ **&**

        **not** $\Gamma_3(x_1, x_2, \ldots, x_n)$ **implies** $\Phi_n(x_1, x_2, \ldots, x_{n-1}, x, r))$
      **holds** $r <> x$;
    **end;**
  **end;**

and similarly in the case of functors defined using the `equals` clause with a simple definiens it is:

  **definition**
    **let** $x_1$ **be** $\theta_1$, $x_2$ **be** $\theta_2$, ..., $x_n$ **be** $\theta_n$;
    **func** $\otimes (x_n)$ -> $\theta_{n+1}$ **equals** *:ident:* $\tau(x_1, x_2, \ldots, x_n)$;
    **coherence;**
    **fixedpoint-free**
    **proof**
      **thus for** $r$ **being** $\theta_{n+1}$, $x$ **being** $\theta_n$ **st** $r = \tau(x_1, x_2, \ldots, x_{n-1}, x)$
        **holds** $r <> x$;
    **end;**
  **end;**

and using the `equals` clause with a complex definiens it is:

  **definition**
    **let** $x_1$ **be** $\theta_1$, $x_2$ **be** $\theta_2$, ..., $x_n$ **be** $\theta_n$;
    **func** $\otimes (x_n)$ -> $\theta_{n+1}$ **equals** *:ident:*
      $\tau_1(x_1, x_2, \ldots, x_n)$ **if** $\Gamma_1(x_1, x_2, \ldots, x_n)$,
      $\tau_2(x_1, x_2, \ldots, x_n)$ **if** $\Gamma_2(x_1, x_2, \ldots, x_n)$,
      $\tau_3(x_1, x_2, \ldots, x_n)$ **if** $\Gamma_3(x_1, x_2, \ldots, x_n)$
      **otherwise** $\tau_n(x_1, x_2, \ldots, x_n)$;
    **coherence;**
    **consistency;**
    **fixedpoint-free**
    **proof**
      **thus for** $r$ **being** $\theta_{n+1}$, $x$ **being** $\theta_n$ **st**
        $(\Gamma_1(x_1, x_2, \ldots, x_n)$ **implies** $r = \tau_1(x_1, x_2, \ldots, x_{n-1}, x, r))$ **&**
        $(\Gamma_2(x_1, x_2, \ldots, x_n)$ **implies** $r = \tau_2(x_1, x_2, \ldots, x_{n-1}, x, r))$ **&**
        $(\Gamma_3(x_1, x_2, \ldots, x_n)$ **implies** $r = \tau_3(x_1, x_2, \ldots, x_{n-1}, x, r))$ **&**
        $(\textbf{not } \Gamma_1(x_1, x_2, \ldots, x_n)$ **& not** $\Gamma_2(x_1, x_2, \ldots, x_n)$ **&**
          **not** $\Gamma_3(x_1, x_2, \ldots, x_n)$ **implies** $r = \tau_n(x_1, x_2, \ldots, x_{n-1}, x, r))$
      **holds** $r <> x$;
    **end;**
  **end;**

This property can also be declared with the `wrt` clause as described in 'Properties', and could be added to Table 1.

An important part of our work was implementing a tool (FIXEDPOINTFREEDETECTOR) which detects MML theorems that could be rewritten as `fixedpoint-free` properties of operations used in formulations of the theorems. To detect such theorems, the following steps should be done:

1. If the theorem is a conjunction of some atomic formulae, decomposing a given formula to a list of atomic formulae.
2. Selecting inequalities among the atomic formulae.
3. Selecting formulae which compare unary terms with single variables among the inequalities.
4. For each such a formula checking whether:
   (a) the argument of the unary term equals to a single variable,
   (b) the type of the variable and the type of the argument of the term declared in the definition of the operation are equal.
5. If both answers to the above questions (4a) and (4b) are positive, marking the fact to be replaceable by the `fixedpoint-free` property of the operation.

At the end of this section we present results of launching the detector on the Mizar Mathematical Library. In the current version of the library 3 such theorems were found in 3 articles. They are: the power set of a set, the successor of a set, and poles at infinity of elements of the absolute. Changing the theorems to the properties caused that 10 inferences in the Mizar Mathematical Library became obvious. Even though these numbers obtained in tests are not too big, the Library Committee of the Mizar project will analyze the cases and will decide about incorporating them into the library. In the case of approval, a refactoring (*Grabowski & Schwarzweller, 2007*) of the MML will be processed.

Computations were carried out at the Computer Center of University of Białystok http://uco.uwb.edu.pl.

## CONCLUSIONS AND FUTURE WORK

Although the basic concept of properties had been introduced to quite early releases of the Mizar system, we still see possibilities to design and develop new features of properties in Mizar. In this paper we described two new features: (a) we presented the syntax and semantics of a new property (`fixedpoint-free`) which enriches the Mizar language and increases the computational power of the Mizar checker by a more automatic processing of unary operations with no fixed points, and (b) we removed a restriction on the application of already defined properties for fixed positions of visible arguments. Investigating a more general approach to introducing properties resulted in an extension of the Mizar language and a corresponding enhancement of the Mizar proof-checker. To analyze the potential usefulness of the proposed general approach we implemented a dedicated software tool and conducted appropriate tests with it on the current Mizar Mathematical Library.

As future work, we plan to open the system of properties for arbitrary (when possible) arities of predicates and functors. We already see within the current content of the Mizar Mathematical Library several applications of that approach, e.g., we would be able to define commutativity for enumerated sets with cardinality greater than two, reflexivity and symmetry of the relation of lying more than two points on a given line, and others. We

also plan to investigate new sorts of properties, like associativity or being one-to-one of functors.

### Funding
The authors received no funding for this work.

### Competing Interests
The authors declare there are no competing interests.

### Author Contributions
- Artur Korniłowicz conceived and designed the experiments, performed the experiments, analyzed the data, performed the computation work, prepared figures and/or tables, authored or reviewed drafts of the paper, and approved the final draft.

### Data Availability
Binary files of the programs and Mizar library required for testing are available at: Artur Korniłowicz. (2020, November 7). Fixedpoint-free property processor. Zenodo. http://doi.org/10.5281/zenodo.4255536.

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
