# Peer review of "Enhancement of properties in Mizar"

_PeerJ Computer Science, doi:10.7717/peerj-cs.320_

## Round 0.1 · original submission · Major Revisions

Please address the comments raised by the reviewers. In particular, I would like you to clarify the overall research objective, and to address the point raised by reviewer 2 about the relation between contribution 1 and 2 and currying.

Reviewer 1 ·

Basic reporting

The language and article format of the manuscript
should be checked to communicate meaning clearly and
use professional English as much as possible.

For example:

Line 54 - rather than "a bunch of computer programs",
perhaps "a collection of computer programs"

Line 94 - "...permissive assumptions are missed" means
"...permissive assumptions are omitted"?

Line 119 - in place of "...variables are permitted what
enables to introduce schemes", "...variables are permitted
which enables the introduction of schemes" or "...variables
are permitted enabling the introduction of schemes" might
be better

Table 1 - table title should be placed above the table

Experimental design

The research objective (or main questions that
need to be answered) should be clarified in the
abstract and paper, especially from a computer science
standpoint. The journal does not accept case studies
or case reports.

Validity of the findings

The author makes a good effort to make the
contents of the paper self-contained, but it
is difficult to link the conclusion to the
research question (which should be revised
and clarified in the experimental design) of
the work.

Reviewer 2 ·

Basic reporting

The paper is written in a clear and understandable form.
Sections 1,2 and 3 give a rich and complete snapshot of the current state of the verification research revolving around the Mizar proof checker.

I only have some minor remarks about style and form, reported below.

In the abstract, the apposition "proof-assistant" explaining what Mizar is only appears at its second occurrence.
It would be better to qualify Mizar at its first occurrence.

'"Property" in Mizar is a construction that can be used to...': reads strange.
What about 'Properties in Mizar are keywords that can be used to...' or
'A property in Mizar is a construction that can be used to...'?

l67: "ways how" -> "how"
l184: The "Next submodule"
l190: "much reacher"
l424: "properties of used in the theorems functors."

Experimental design

Section 3 provides a detailed background of the role of properties in the Mizar language and of their current limitations.
This allows the author to clearly point out what the paper contributes.
Again, I only have minor remarks about some points where there possibly is some technicalities excess.

The word "functor" has usually a different meaning in mathematics and theoretical computer science, coming from category theory. Perhaps the reader should be warned about this.
There are other instances where specific Mizar jargon is used and where some reference or explanation would benefit the reader. For example, l71: permissive assumption.

line 30: to satisfy the interested reader, it would be better to have a reference which uses identifications or explain what they are.

Validity of the findings

The contributions of the paper are:
1) a generalisation of the existing properties feature of Mizar allowing to apply them to functions/operations of arbitrary arities;
2) correspondingly, an optional, additional "wrt" keyword allowing the user to precisely specify which arguments of the function/operation are affected by the property;
3) a new property (fixpoint-free)

1 and 2 are formally described in section 3, and 3) in section 4.
Wrt 3), some small investigation about its impact on MML is proposed.
I am surprised that only 3 occurrences have been found; I don't think this means the new property is useless.
On the contrary, I suspect many more occurrences can be found, making it useful.
There are many numerical functions satisfying this property. Even not taking into account functions whose output type differs from their arguments' types, which should automatically satisfy that property.

Regarding 1) and 2), section 3.2 is important for researchers working within (as opposed to merely using) theorem proving.

If I may suggest a possible line of further investigation: 1) and 2) seem intimately related to the notion of currying, which I understand is defined in MML. Would that be possible to link the two things together? If yes (and I am not sure, given the first-order nature of Mizar!), how?

The small example at the end of section 3.1 is intriguing: I understand that some properties are not enforceable at definition time because they only apply to a subclass of the defined items. However, is there a workaround in such cases, maybe based on cluster registrations?

---

## Round 0.2 · accepted · Accept

As you can see, the reviewer was very happy with your changes.

Reviewer 2 ·

Basic reporting

The author addressed my points in a satisfactory manner.
Therefore, I am happy to confirm the positive overall assessment of the paper expressed in my previous review.

Experimental design

The author addressed my points in a satisfactory manner.
Therefore, I am happy to confirm the positive overall assessment of the paper expressed in my previous review.

Validity of the findings

The author addressed my points in a satisfactory manner. In particular, I am happy with the technical explanation addressing the question about currying.
Therefore, I am happy to confirm the positive overall assessment of the paper expressed in my previous review.

Additional comments

The diff paper version and the colour-coded rebuttal eased my job: I would like to thank the author for the appreciated effort.